# vFedSec: Efficient Secure Aggregation for Vertical Federated Learning via Secure Layer

## Abstract

Most work in privacy-preserving federated learning (FL) has been focusing on horizontally partitioned datasets where clients share the same sets of features and can train complete models independently. However, in many interesting problems, individual data points are scattered across different clients/organizations in a vertical setting. Solutions for this type of FL require the exchange of intermediate outputs and gradients between participants, posing a potential risk of privacy leakage when privacy and security concerns are not considered. In this work, we present *vFedSec* - a novel design with an innovative *Secure Layer* for training vertical FL securely and efficiently using state-of-the-art security modules in secure aggregation. We theoretically demonstrate that our method does not impact the training performance while protecting private data effectively. Empirical results from extensive experiments substantiate this design producing secure training with negligible computation and communication overhead. Compared to widely-adopted homomorphic encryption (HE) methods, our method can obtain $\geq 690\times$ speedup and reduce communication costs by $\geq 9.6\times$.

## 1 Introduction

Federated Learning (FL) is a machine learning paradigm that enables the training of a global model using decentralized datasets without requiring the sharing of raw and sensitive data from participating parties (McMahan et al., 2017; Li et al., 2020; Yang et al., 2019). Under FL, individual institutions or devices train a common global model collaboratively, agreeing on a central server to orchestrate the learning and perform model aggregation.

In terms of data partitioning, FL can be categorized as either horizontal or vertical scenarios. Most existing works focus on horizontal FL (HFL), which requires all participants to use the same feature space but different sample spaces(Yang et al., 2019). This partitioning scheme is usually found in the cross-device setup where clients are often mobile or IoT devices with heterogeneous datasets and resources under complex distributed networks (Yu & Li, 2021; Qiu et al., 2022). Under Vertical FL (VFL) (Wei et al., 2022), however, data points are partitioned across clients, which means different clients might hold different features for the same sample. This is often found in cross-silo setups where participating clients, such as hospitals and research institutions, may hold complementary pieces of information for the same data points. For example, two different hospitals may hold different clinical data for the same patient.

The need for VFL has arisen massively in the industry these years (Liu et al., 2022; 2020). For example, a financial institution would like to train a financial crime detection model, but it only has limited features in its own institution that restrict performance. The institution would like to have access to more private information, such as account information, that various banks might have. However, this is a deal breaker for financial applications as such data is very sensitive, and legal restrictions (e.g., GDPR) can prevent it from being shared across institutions. Another example will be in the commercial ad ranking systems, in the sense that each organization might have different information for the same customer, but the label (the click rate) will only be stored in the application platform. With VFL, institutions and companies that own only small and fragmented data have constantly been looking for other institutions to collaboratively develop a shared model for maximizing data utilization (Li et al., 2021b).

Due to their different data structures, training procedures for HFL and VFL can be very different. Each client in HFL trains a complete copy of the global model on their local dataset and sends model updates to a centralized server for aggregation. Under VFL, each client, holding certain features of the whole dataset, contributes to a sub-module of the global model. This means that intermediate activations or gradients need to be shared between clients during the training process, posing a potential risk for privacy leakage, as the original raw data can be reconstructed from said gradients (Zhu et al., 2019; Zhao et al., 2020; Jin et al., 2021; Yin et al., 2021). While most research has focused on designing methods to train the global model under VFL better, fewer efforts have been devoted to providing a secure way of training.

In this work, we present *vFedSec* - an efficient and privacy-preserving way of training under the Vertical FL setup. Section 4 describes our proposed design and introduces its core component, the architecture-agnostic *Secure Layer*. We provide a robust theoretical justification showing that our proposed method does not adversely affect training performance and private information is protected through the Secure Layer. To validate our claims empirically, we have conducted comprehensive experiments over four varied datasets of differing data volumes and model architectures, as detailed in Section 6. The results substantiate that our vFedSec imposes negligible overhead in all cases. Moreover, our method demonstrates a remarkable $\geq$ 690x speedup and $\geq$ 9.6x decrease in communication costs when compared to the resource-intensive homomorphic encryption (HE) techniques, thus affirming its superior computational and communication efficiency.

## 2 BACKGROUND AND RELATED WORK

### 2.1 GENERAL VERTICAL FEDERATED LEARNING

**We consider a general form of Vertical Federated Learning, where data are partitioned both in the sample and feature spaces.** Specifically, we investigate $C$ class classification problem defined over a compact feature space $\mathcal{X}$ and a label space $\mathcal{Y} = [C]$, where $[L] = \{1, ..., C\}$. Since each client only holds a subset of the whole feature space, we can divide the feature into the finite number of sub-spaces $\{\mathcal{X}_1, ..., \mathcal{X}_n\}$, and each client holds feature from one sub-space. Following the setup as in previous literature (Liu et al., 2022), we define two kinds of clients in the VFL settings. The first type is the *active party* ($ClientA$), which holds all the samples with the ground-truth labels and multiple features, and there usually is only one active party. The second type is called the *passive parties*, which only holds some other features.

All clients can be clustered by the feature set they own. If they hold the features from $\mathcal{X}_i$, we say these are from client cluster $i$. Multiple clients can hold different samples with the feature from the same feature sub-space. Let $f$ be the function for the neural network parameterized over the hypothesis class $w$, which is the weight of the neural network. $\mathcal{L}(\mathbf{w})$ is the loss function, and we assume the widely used cross-entropy loss.

In a horizontal FL scenario, where datasets are horizontally partitioned, FL solutions are typically derived from a centralized one, providing an upper-bound target efficiency. However, this approach is not well suited in the case of vertically-partitioned FL, where different feature types are distributed across different participants.

Training in a vertical FL setting consists of two parts. The first part is the local module, which outputs an intermediate output ($H$). The local module can be embedding layers to extract embedding from each feature. The second part is the global module hosted by the server, which takes the embeddings from each client and further propagates to make the final predictions. It is worth noticing that if two clients are from the same client cluster, they will have the same architecture of the local module, which means that their local module can be updated via the horizontal FL way, and use aggregation methods such as FedAvg (McMahan et al., 2017).

### 2.2 RELATED WORK

In the context of efficient secure aggregation for VFL, several related papers have contributed to the understanding and improvement of privacy, utility, and security aspects in VFL. A comprehensive literature review (Liu et al., 2022) provides an overview of VFL and discusses various protection algorithms. However, it does not specifically address efficient secure aggregation. Another study (Ranbaduge & Ding, 2022) focuses on integrating differential privacy (DP) techniques into VFL, but it

does not explicitly cover efficient secure aggregation either. Kang et al. (2022b) propose a framework to evaluate the privacy-utility trade-off in VFL, highlighting the limitations of existing approaches like Homomorphic Encryption (HE) and Multi-party Computation (MPC) in terms of computation and communication overheads, and Cai et al. (2023) apply HE to Split Neural Networks (Split NN) for improved security. Zheng et al. (2022) presents a heuristic approach to make Split Learning resilient to label leakage, albeit at the cost of accuracy. Sun et al. (2022) explore label leakages from forward embeddings and their associated protection methods, which can cause a decrease in accuracy. Although these papers contribute to the understanding of VFL, there is a research gap in efficient secure aggregation techniques for VFL that needs to be addressed. The method proposed by Liu et al. (2020) only tries to protect the sample IDs, rather than all the raw private data; Chen et al. (2020) perturbed local embedding to ensure data privacy and improve communication efficiency, which has strict requirements for the embedding and can impact the overall performance. There are also BlindFL (Fu et al., 2022), ACML (Zhang & Zhu, 2020), and PrADA (Kang et al., 2022a) which are all homomorphic encryption (HE) based solutions. These approaches often incur significant communication and computation overheads. Moreover, their fixed design cannot be extended to multiple-party scenarios.

# 3 SECURE LAYER: SECURE AGGREGATION VIA MASKING WITH NOISE

The core of *vFedSec* resides on the Secure Layer, which combines the intermediate output from each client using secure aggregation. We would like to emphasize that the "Secure Layer" is an abstraction of the foundational mechanisms inherent in our algorithm. It is a conceptual layer, whose computation is completed by the collaboration of different parties. Secure Layer is essentially a fully connected layer in terms of model architecture, which computes its output $Z$ as follows.

$$Z = WH = (W_1, W_2, \ldots, W_n)(H_1, H_2, \ldots, H_n)^\mathsf{T} = \sum_{i=1}^{n} W_i H_i \tag{1}$$

where $W$ is the layer parameters, $H$ is the concatenated intermediate output from clients, and $n$ is the number of clusters.

As introduced in Section 2.1, clients can be grouped into clusters, within each of which any two hold the same set of features but different samples. Therefore, clients (passive parties) in one cluster share the same local module. Client A, the active party, is in a cluster containing only itself. After Client A selects a mini-batch, the client in cluster $i$ that holds the sample will produce batched embeddings $H_i$ (see Equation 1). To be specific, let $\{x^{(1)}, x^{(2)}, \ldots, x^{(B)}\}$ and $f_i$ denote the selected batch and the local module of cluster $i$ respectively. As samples are distributed among the cluster, each client $p$ in the cluster $i$ will compute the embeddings of samples owned by itself and fill other embeddings with zeros, i.e., $[H_i]_p = \left(\mathbb{1}(x^{(k)} \in D_p) f_i(x^{(k)})\right)_{k=1}^{B}$, where $D_p$ is the local data held by client $p$. The output of the Secure Layer can then be computed by the following equation.

$$Z = \sum_{i=1}^{n} W_i H_i = \sum_{i=1}^{n} \sum_{p \in \mathcal{C}_i} W_i [H_i]_p \tag{2}$$

where $\mathcal{C}_i$ is the cluster $i$.

The Secure Layer enables each client to mask each intermediate output with noise, hence hiding any client's confidential information from any other party. Following the idea of *Secure Aggregation*, we make added noises cancel out each other, i.e., $\sum_p \mathbf{n}_p = \mathbf{0}$, which is the summation of a series of random numbers generated by a Pseudo-Random Generator (PRG) that can generate sequences of uniformly pseudo-random numbers given a seed.

The seed is the pairwise shared secrets $ss_{ij} = ss_{ji}$ established in the Setup phase (Section 4.1). The added noise for each client $i$ is then generated as shown in Equation 3 and 4. Here, we treat the active party Client $A$ as Client 1. It is worth noticing that for Client $i$, the noise is calculated by adding all the random numbers generated by the client with ID above $i$ and minus the random number generated by the client with ID below $i$. Therefore, in this way, the summation of all the noise can be canceled out, resulting in the overall summation of 0.

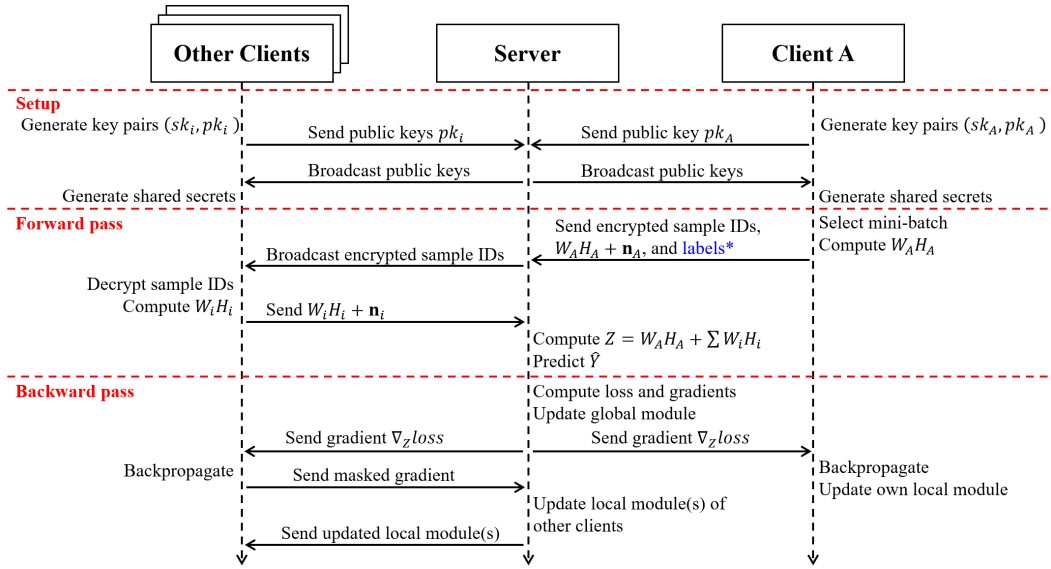

Figure 1: A high-level view of our protocol. Labels in the forward pass section are only required in training. Though each client only generates one key pair as in the plot, it is recommended in practice to generate at least two key pairs for noise generation and symmetric encryption to mitigate the risks of severe data breaches in the event of unexpected secret keys or shared secrets leakage.

$$\mathbf{n}_i = -\sum_{j<i} \mathrm{PRG}(ss_{ij}) + \sum_{j>i} \mathrm{PRG}(ss_{ij}) \tag{3}$$

$$\sum_i \mathbf{n}_i = \sum_i \sum_{j>i} \big( \mathrm{PRG}(ss_{ij}) - \mathrm{PRG}(ss_{ji}) \big) = \mathbf{0} \tag{4}$$

In each forward pass, the server will broadcast the respective weight ($W_i$) of the Secure Layer to each client, and then each client $p$ in cluster $i$ can generate noise $\mathbf{n}_p$ and return the masked intermediate output $W_i[H_i]_p + \mathbf{n}_p$. After receiving all masked intermediate output from clients, the server can compute output accurately without knowledge of any individual value from each client by adding them together, i.e., $Z = \sum W_i[H_i]_p + \sum_p \mathbf{n}_p = WH$, as shown in Equation 1 and 2.

As proven in Section 5.1, the added noise ensures that the masked intermediate output is indistinguishable from a uniformly random sequence, thus providing strong privacy guarantees. An adversary can unmask an individual intermediate output of a party only if it knows all the shared secrets held by that party. In other words, Secure Layer can guarantee the confidentiality of the intermediate output of any client unless all other clients are corrupted. Due to the nature of PRG, as long as the seed of PRG is not reset, the random number sequences generated by PRG in different rounds will be different, guaranteeing that any other party cannot recover the noise without having access to the seed. In practice, participating parties should periodically generate new shared secrets in case of leakage, e.g., creating new shared secrets every 5 training rounds.

## 4 VFEDSEC WITH SECURE LAYER

In this section, we propose our novel design *vFedSec* for VFL to ensure the privacy and security of sensitive information in local datasets. We summarize our training protocol in Figure 1.

Our protocol is divided into three phase: *setup phase* (Section 4.1), *training phase* (Section 4.2), and *testing phase* (Section 4.3). The setup phase is designed for configuring training procedures, including key exchange, mini-batch selection, forward pass, and backward pass. Since features are distributed across different clients, both the forward pass and the backward pass cannot be computed at the same place at the same time. Therefore, we propose our novel design of *Secure Layer*, which will be explained in detail in Section 3.

We also show that our design hides the sensitive information completely as we use encryption to help the mini-batch selection *without revealing identifiable information to any third party*. In addition, we adapt the idea of *secure aggregation via masking with noise* (Bonawitz et al., 2016) and introduce an innovative *Secure Layer* for the activations and gradients aggregation.

We design the protocol to be able to cope with any kind of VFL setup and any kind of model architectures reside both in the local module on the client side and the global module on the server side. As explained in Section 2.1, our setup is a generalized VFL setup that includes all possible variations and scenarios in the area of VFL.

## 4.1 SETUP PHASE

The first step of *vFedSec* is to generate shared secrets. We use the Elliptic-curve Diffie-Hellman (ECDH) key agreement protocol (Barker et al., 2017; Diffie & Hellman, 1976) to generate shared secrets through insecure channels between all clients. The shared secrets will be used to build secure pairwise channels by symmetric encryption and facilitate secure aggregation. During the setup phase, the central aggregator requests public keys from all participating clients. Then, $\forall i$, Client $i$ generates one pair of secret key $sk_i^{(j)}$ and public key $pk_i^{(j)}$ for each Client $j$ and sends public keys to the aggregator. $\forall i \neq j, pk_i^{(j)}$ is then forwarded to Client $j$. Once received, Client $i$ and Client $j$ can generate a shared secret $ss_{ij} = ss_{ji}$ from $(sk_i^{(j)}, pk_j^{(i)})$ or $(sk_j^{(i)}, pk_i^{(j)})$.

## 4.2 TRAINING PHASE

Similar to standard FL training, the server will initialize the initial model parameters $w_0$ and broadcast the respective model weights to the relevant parties. It is worth noting that the local module is still initialized from the server, since local modules from the same client cluster will have the initialization.

**Mini-batch selection**   We assume that the active party knows which passive parties hold the features of a given sample. This can be realized by Private Set Intersection (Lu & Ding, 2020; Zhou et al., 2021). We denote the identifier for samples as sample ID, which is shared among all parties. Since the active party has the information regarding each sample and the ground truth label, the mini-batch selection will start from the active party. It will first select a batch of data in the active party ($Client\ A$). The sample ID will be encrypted using $ss_{Ai}$ as key if the partial features of the sample are held by passive party $Client\ i$. The active party will then upload the encrypted ID batch to the aggregator, which will broadcast it to all passive parties. As sample IDs are encrypted using different keys, each passive party can only decrypt sample IDs existing in its dataset, which prevents any party from knowing extra information about the batch and samples in it.

**Forward Pass**   After the local module in each client, each client will have intermediate output $H_i$, which is then fed into the Secure Layer as explained in detail in Section 3. Then, the output of the Secure Layer is passed through the global module to finish the forward pass.

**Backward Pass**   After the mini-batch selection, the active party will then share the ground-truth label in the order of the mini-batch to the server. We believe it is reasonable to do this as the server cannot infer any useful information from only the label. Regarding the backward pass through the Secure Layer, the gradient of the Secure Layer ($\nabla_Z \mathcal{L}$) can be computed and sent to the active party, which then can finish the back-propagation and update the local module in the active party.

Regarding the local module in the passive parties, similar to the active party, the gradient of the Secure Layer ($\nabla_Z \mathcal{L}$) can be computed and sent to all the passive parties, which then can compute the gradient of their local module. The gradient of the local module is hidden by adding pair-wise masks among clients in the same cluster ($\mathcal{C}$). Similar to the Secure Layer (see Equation 3 4), the masks are constructed such that their summation will be canceled out.

Then, the passive parties send the masked gradient to the server, which will then aggregate the gradient and obtain the averaged gradient for the mini-batch $g = 1/n \sum_i \nabla \mathcal{L}_i + \mathbf{n}_i = 1/n \sum_i \nabla \mathcal{L}_i$. The averaged gradient can then be used to update the local module for each client cluster. If there is only one client in the cluster, the local module can be updated in the same way as the active party.

### 4.3 TESTING PHASE

During the testing phase, the active party will first send the encrypted batch information similar to the training phase to the aggregator. After receiving the encrypted batch information, the encrypted batch information is shared with the passive clients. Then the final prediction is computed the same way as the forward pass in the training phase utilizing the Secure Layer.

## 5 SECURITY ANALYSIS

### 5.1 THREAT-MODEL AND PRIVACY GUARANTEE

We consider a *threat model* where both clients and the server are *honest-but-curious*, i.e., they are expected to follow the pre-defined training protocol whilst trying to learn as much information as possible from the models, intermediate output, or gradients that they receive. In our security arguments, we will use the following Lemma (Bonawitz et al., 2016). Proof for the lemma can be found in Appendix A.2.

**Lemma 5.1.** Fix $n$, $m$, $R$, $U$ with $|U| = N$, and $\{x_u\}_{u \in U}$ where $\forall u \in U, x_u \in \mathbb{Z}_R^m$. Then,

$$\{\{p_{u,v} \xleftarrow{\$} \mathbb{Z}_R^m\}_{u<v},\ p_{u,v} = -p_{v,u} \forall u > v : \{x_u + \sum_{v \in U \setminus \{u\}} p_{u,v} \ (\mathrm{mod}\ R)\}_{u \in U}\}$$

$$\equiv$$

$$\{\{w_u \xleftarrow{\$} \mathbb{Z}_R^m\}_{u \in U} s.t. \sum_{u \in U} w_u = \sum_{u \in U} x_u (\mathrm{mod}\ R) : \{w_u\}_{u \in U}\}$$

where '$\equiv$' denotes that the distributions are identical, $\xleftarrow{\$}$ denotes uniform sampling, $R$ is the upper bound of the integer set $\mathbb{Z}_R^m = \{0, 1, \ldots, R-1\}^m$

Here $U$ can be seen as a set of clients in our protocol. The lemma proves that if clients' values have uniformly random pairwise masks added to them, then the resulting values look uniformly random, conditioned on their sum being equal to the sum of the client's values. In other words, the pairwise masks as explained in Section 3 hide all private information about the client's data, apart from the sum. Secure aggregation through *Secure Layer* is achieved through the use of masking on the intermediate outputs and gradients before they are sent back to the server, thus preventing the server from using the received information to gain knowledge about the sensitive client data. Therefore, our method protects the data from both data reconstruction and membership inference attacks.

Also, while our current implementation could be vulnerable to active adversaries, our solution can be extrapolated very easily to include *malicious* settings by introducing a public-key infrastructure (PKI) that can verify the identity of the sender (Bonawitz et al., 2017). It can thus be further protected from malicious attacks.

Although our method does not allow exposing secret keys to other parties and demonstrates robustness against collusion between the aggregator and passive parties, in practical settings, the risk of secret key leakage persists, e.g., accidental inappropriate practices of engineers. Thus, to ensure privacy, it is necessary to routinely regenerate keys for symmetric encryption and secure aggregation, specifically, by executing the setup phase after every K iteration, in both the training and testing stages. The value of K can vary in real-world scenarios, but the larger value will inevitably incur higher risks of keys being compromised. In the event of key leakage, an attacker will only have access to a limited amount of information instead of all encrypted information if keys are regenerated periodically.

### 5.2 OTHER CONSIDERATIONS

**Generalizability:** *vFedSec* provides a generalized protocol for any kind of VFL setups as explained in Section 2.1. We design our protocol *vFedSec* to be able to cope with any kind of vertical FL setups and any kind of model architectures residing both in the local module on the client side and the global module on the server side. Previous papers usually only consider the vertical setup with two

parties: one active and one passive (Liu et al., 2022), or the feature in passive parties cannot overlap (Jiang et al., 2022; Fang et al., 2021; Wei et al., 2022). However, in our general setup, there could be multiple passive parties, and the feature space in passive parties can be overlapped, as described in the client cluster in Section 2.1. Also, our protocol can be further expanded to include multiple active parties if needed. In this case, we just need to adapt the mini-batch selection process and the label-sharing process with the server.

**Scalability and Efficiency:** Our core solution is agnostic to the number of participating clients and the data partition schemes under the VFL setting. As a result, our solution's scalability is only dependent on the underlying FL framework and on how key generation and key exchange between clients are handled. Also, our solution is scalable and efficient in the sense that, unlike many previous methods explained in Section 2.2 that utilized HE, we employ lightweight masks through random noise, and the mask can be naturally decrypted through summation.

**Dropout:** In the realm of VFL, several studies have delved into its security and robustness. Gu et al. (2021) presented AFSGD-VP, tailored for environments without a central server, ensuring embedded privacy through a tree-structured aggregation. Yan et al. (2022) introduced AMVFL, which focuses on asynchronous gradient aggregation for linear and logistic regression, with local embeddings shielded by secret-shared masks. VAFL (Chen et al., 2020) is designed for intermittently connected clients, incorporating Differential Privacy (DP) but with some performance compromises compared to our vFedSec. Notably, FedVS (Li et al., 2023) utilizes polynomial interpolation to efficiently handle stragglers, yet it falls short in addressing the dropout challenge. More importantly, Since each client might contribute to a particular sub-module of the whole model, client dropout can lead to missing features, hence impacting the overall training (Fu et al., 2022; Zhang & Zhu, 2020; Kang et al., 2022a). Therefore, we consider the case with no client drop out, which is a realistic assumption given the nature of VFL.

## 6 EXPERIMENTS

We conducted extensive experiments on five classification datasets with various model architectures. Federated learning is simulated using the FL framework Flower (Beutel et al., 2020).

### 6.1 DATASETS

Experiments are conducted over five datasets: *Banking dataset* (Moro et al., 2011), *Adult income dataset* (Kohavi et al., 1996), *Taobao ad-display/click dataset* (Li et al., 2021a), EMNIST (Cohen et al., 2017), and FMNIST (Xiao et al., 2017). The banking dataset is related to the direct marketing campaigns of a Portuguese banking institution. It contains $45,211$ rows and $18$ columns ordered by date. The adult income dataset is a classification dataset aiming to predict whether the income exceeds 50K a year based on census data. It contains $48,842$ and $14$ columns. We also conduct our experiment over a production scale ad-display/click dataset of Taobao (Li et al., 2021a). The dataset contains 26 million interactions (click/non-click when an Ad was shown) and $847$ thousand items across an 8-day period. The EMNIST balanced dataset is an expansion of the original MNIST to include handwritten characters, which contains $131,600$ images from 47 balanced classes. The FMNIST dataset, i.e., the Fashion MNIST dataset, comprises $70,000$ grayscale images of clothing items and accessories.

The specific dataset partition, model architecture, and the details of overhead profiling can be found in Appendix A.1

### 6.2 RESULTS

We conduct experiments over five datasets to measure both the computation and the communication cost of *vFedSec* training. The computation cost is measured through CPU time (in seconds), and the communication cost is measured through the transmission size (in MB). We also measure the overhead cost that shows the extra CPU time or communication compared to unsecured VFL training. Communication overheads are from the length of ciphertext (encrypted Sample IDs) over plaintext and the exchange of public keys. Computation overheads are caused by creating key pairs and shared secrets, noise generation and cancellation, and encryption and decryption. As transferring parameters/gradients and updating models are necessary for VFL training and split learning, they are

| | Active Party CPU time (s) | | | | Passive Party CPU time (s) | | | |
|---|---|---|---|---|---|---|---|---|
| | Training phase | | Testing phase | | Training phase | | Testing phase | |
| Dataset | Total | Overhead | Total | Overhead | Total | Overhead | Total | Overhead |
| **Banking(Ours)** | **1.044** | **0.1881** | **0.3057** | **0.1864** | **0.1633** | **0.1089** | **0.1363** | **0.1075** |
| Banking(SEAL) | $1.075 \times 10^3$ | $1.075 \times 10^3$ | $1.075 \times 10^3$ | $1.075 \times 10^3$ | $1.113 \times 10^2$ | $1.113 \times 10^2$ | $1.113 \times 10^2$ | $1.113 \times 10^2$ |
| Banking(PHE) | $1.565 \times 10^4$ | $1.565 \times 10^4$ | $1.565 \times 10^4$ | $1.565 \times 10^4$ | $1.993 \times 10^3$ | $1.993 \times 10^3$ | $1.993 \times 10^3$ | $1.993 \times 10^3$ |
| **Income(Ours)** | **0.7379** | **0.1909** | **0.2738** | **0.1889** | **0.1588** | **0.1143** | **0.1413** | **0.1124** |
| Income(SEAL) | $5.093 \times 10^2$ | $5.092 \times 10^2$ | $5.093 \times 10^2$ | $5.092 \times 10^2$ | $3.885 \times 10^2$ | $3.884 \times 10^2$ | $3.884 \times 10^2$ | $3.884 \times 10^2$ |
| Income(PHE) | $7.413 \times 10^3$ | $7.413 \times 10^3$ | $7.413 \times 10^3$ | $7.413 \times 10^3$ | $6.853 \times 10^3$ | $6.853 \times 10^3$ | $6.853 \times 10^3$ | $6.853 \times 10^3$ |
| **Taobao(Ours)** | **1.836** | **0.1767** | **0.4085** | **0.1751** | **0.1403** | **0.1037** | **0.1222** | **0.1022** |
| Taobao(SEAL) | $7.429 \times 10^3$ | $7.428 \times 10^3$ | $7.429 \times 10^3$ | $7.428 \times 10^3$ | $1.631 \times 10^2$ | $1.631 \times 10^2$ | $1.631 \times 10^2$ | $1.631 \times 10^2$ |
| Taobao(PHE) | $1.082 \times 10^5$ | $1.082 \times 10^5$ | $1.082 \times 10^5$ | $1.082 \times 10^5$ | $2.945 \times 10^3$ | $2.945 \times 10^3$ | $2.945 \times 10^3$ | $2.945 \times 10^3$ |
| **EMNIST(Ours)** | **1.040** | **0.3334** | **0.6671** | **0.3300** | **0.8538** | **0.2314** | **0.5386** | **0.2314** |
| EMNIST(SEAL) | $7.918 \times 10^4$ | $7.918 \times 10^4$ | $7.918 \times 10^4$ | $7.918 \times 10^4$ | $7.918 \times 10^4$ | $7.918 \times 10^4$ | $7.918 \times 10^4$ | $7.918 \times 10^4$ |
| EMNIST(PHE) | $1.153 \times 10^6$ | $1.153 \times 10^6$ | $1.153 \times 10^6$ | $1.153 \times 10^6$ | $1.153 \times 10^6$ | $1.153 \times 10^6$ | $1.153 \times 10^6$ | $1.153 \times 10^6$ |
| **FMNIST(Ours)** | **1.016** | **0.3240** | **0.6286** | **0.3189** | **0.8504** | **0.2301** | **0.5282** | **0.2300** |
| FMNIST(SEAL) | $7.918 \times 10^4$ | $7.918 \times 10^4$ | $7.918 \times 10^4$ | $7.918 \times 10^4$ | $7.918 \times 10^4$ | $7.918 \times 10^4$ | $7.918 \times 10^4$ | $7.918 \times 10^4$ |
| FMNIST(PHE) | $1.153 \times 10^6$ | $1.153 \times 10^6$ | $1.153 \times 10^6$ | $1.153 \times 10^6$ | $1.153 \times 10^6$ | $1.153 \times 10^6$ | $1.153 \times 10^6$ | $1.153 \times 10^6$ |

Table 1: Results on the CPU time (in seconds). Note that PHE package is less optimized than SEAL. It is generally not considered in real-world applications.

| | Active Party Data Transmission (MB) | | | | Passive Party Data Transmission (MB) | | | |
|---|---|---|---|---|---|---|---|---|
| | Training phase | | Testing phase | | Training phase | | Testing phase | |
| Dataset | Total | Overhead | Total | Overhead | Total | Overhead | Total | Overhead |
| **Banking(Ours)** | **0.88** | **0.14** | **0.57** | **0.14** | **0.79** | **0.13** | **0.44** | **0.13** |
| Banking(SEAL) | 8.41 | 7.67 | 8.10 | 7.67 | 8.32 | 7.57 | 7.97 | 7.57 |
| Banking(PHE) | 68.70 | 67.95 | 68.38 | 67.95 | 68.60 | 67.85 | 68.26 | 67.85 |
| **Income(Ours)** | **0.88** | **0.14** | **0.57** | **0.14** | **0.85** | **0.13** | **0.44** | **0.13** |
| Income(SEAL) | 8.41 | 7.67 | 8.10 | 7.67 | 8.39 | 7.57 | 7.97 | 7.57 |
| Income(PHE) | 68.70 | 67.95 | 68.38 | 67.95 | 68.67 | 67.85 | 68.26 | 67.85 |
| **Taobao(Ours)** | **1.51** | **0.14** | **0.88** | **0.14** | **1.42** | **0.13** | **0.76** | **0.13** |
| Taobao(SEAL) | 16.57 | 15.20 | 15.95 | 15.20 | 16.49 | 15.10 | 15.82 | 15.10 |
| Taobao(PHE) | 137.13 | 135.76 | 136.51 | 135.76 | 137.05 | 135.66 | 136.38 | 135.66 |
| **EMNIST(Ours)** | **6.24** | **0.21** | **3.31** | **0.21** | **5.99** | **0.05** | **3.06** | **0.05** |
| EMNIST(SEAL) | 76.85 | 70.82 | 73.92 | 70.82 | 76.60 | 70.66 | 73.67 | 70.66 |
| EMNIST(PHE) | 641.98 | 635.95 | 639.05 | 635.95 | 641.73 | 635.79 | 638.80 | 635.79 |
| **FMNIST(Ours)** | **6.24** | **0.21** | **3.31** | **0.21** | **5.99** | **0.05** | **3.06** | **0.05** |
| FMNIST(SEAL) | 76.85 | 70.82 | 73.92 | 70.82 | 76.60 | 70.66 | 73.67 | 70.66 |
| FMNIST(PHE) | 641.98 | 635.95 | 639.05 | 635.95 | 641.73 | 635.79 | 638.80 | 635.79 |

Table 2: Results on the communication both in size (MB).

not counted as overheads. All experiments are reported with 1 setup phase and 5 training rounds, and each experiment is repeated 10 times, and averages are reported.

As mentioned in Section 5.1, in practice, each party should create new key pairs routinely to mitigate the risk of adversaries from accessing confidential information in the event of secret key leakage. In our experiments, the key pairs and the shared secrets will be regenerated for every 5 iterations.

In Table 1 we report the CPU time as a measure to show the computation cost using *vFedSec*. The equivalence of certain total times to overhead times can be attributed to the substantial overhead of HE, which overshadows other computational processes. The CPU time is reported separately for the active party and passive parties. The overhead columns show extra CPU time compared with unsecured VFL training. We use HE functions from Python module *Phe* (Data61, 2013) and *SEAL-Python* (Huelse, 2023). *Phe* module implements the Pallier cryptosystem in Python, and *SEAL-Python* creates Python bindings for APIs in Microsoft SEAL (SEAL) using Pybind11 (Jakob et al., 2017). Considering that the protocols proposed by many HE VFL papers, such as BlindFL(Fu et al., 2022), ACML(Zhang & Zhu, 2020), and PrADA(Kang et al., 2022a), they entail encrypting a matrix and computing matrix multiplication between an encrypted matrix and a plaintext matrix, we estimated the overhead of above operations through (1) encrypt a weight matrix in each aggregation; (2) compute a matmul between a plaintext matrix and an encrypted weight matrix in each aggregation. It is worth noting that, in reality, most HE VFL protocols require multiple rounds of communication, repetitively encrypt/decrypt matrices, and do matmul. But, in our estimated cases, we only do encrypt

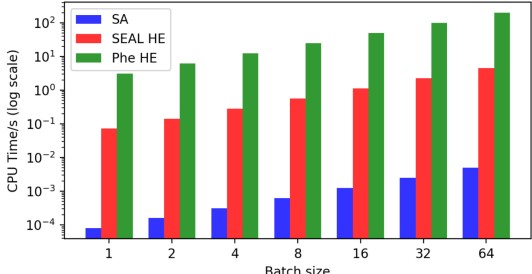

Figure 2: Comparison of average CPU time for different batch sizes, using SA and HE from Phe and SEAL-Python. Y-axis is in log scale. The results are from 10 experiments.

Table 3: Overhead summary of *vFedSec* in one training/testing round. $h$ is the size of intermediate output, $B$ is the batch size, and $N$ is the number of clients.

|  | **Client** | **Server** |
|---|---|---|
| **Computation** | $O(hN + B)$ | $O(1)$ |
| **Communication** | $O(N + B)$ | $O(BN)$ |
| **Storage** | $O(N)$ | $O(1)$ |

and do matmul once in each forward pass, and exclude the overhead of decryption (as decryption may happen on the server side). Thus, our estimation is the minimum overhead incurred by HE VFL protocols, while the real overheads of adopting those protocols can be multiple times larger than our estimation if using the same HE libraries. Table 2 shows the transmission size in bytes for the method and is also demonstrated on both the active and passive parties. As demonstrated in both tables, the overhead accounts for a relatively small part of the total amount, in both CPU time and communication size. The CPU time overhead is caused by parties adding masks to their original output and encryption/decryption of sample IDs. The communication overhead is introduced by broadcasting encrypted sample IDs, which are larger than plain text. As the masks can be cancelled out by summing them, the unmasking process is very efficient.

Additionally, as reported in Table 3, the client-side overheads of *vFedSec* grow linearly with increasing the number of clients and the batch size. The server-side computation and storage overheads are negligible, and the communication overheads are incurred by broadcasting encrypted sample IDs to all clients, instead of only forwarding sample IDs to the sample holders. More results can be found in Appendix A.3.

### 6.3 ABLATION STUDY

We demonstrate the efficiency of *vFedSec* through an ablation study to compare our method and widely-adopted method HE. The experiments compare how *vFedSec* and HE process matrix multiplications.

Assume the input tensor is of size (Batch size, 8), and the weight tenor is (8, 8). Tensor shapes in the comparison are smaller than tensors used by a passive party in experiments. Given that the HE libraries do not support matrix operations, both *vSecFed* and HE implementations are not optimized by any Python modules, such as *numpy*. The HE implementation in this comparison inevitably involves nested Python loops. For large matrices or more advanced operations, implementations in other languages, e.g., C++, C#, are more suitable. The results can be found in Fig. 2, which clearly shows the efficiency of our method. It indicates that our approach (SA) yields a $9.1 \times 10^2 \sim 3.8 \times 10^4$ times speedup when evaluating the computational overheads attributed solely to SA in contrast to HE.

### 7 CONCLUSION

In this work, we consider the challenge of privacy-preserving training in vertical federated learning settings. We provide the first framework termed *vFedSec* to use secure aggregation in the setting of vertical FL by introducing innovative *Secure Layer*, which effectively and efficiently protects the private sensitive dataset by masking the activations and gradients during aggregation. Our method is efficient and accurate in the sense that it will not change the underlying results and performance by adding security modules. We also benchmark our method against HE-based solutions using two HE libraries, which indicate a reduction in computational costs by at least $690\times$, and a more than $9.6\times$ decrease in communication costs.

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

# A   APPENDIX

## A.1   EXPERIMENT SETUP

**Banking Datasets:** We keep the `housing`, `loan`, `contact`, `day`, `month`, `campaign`, `pdays`, `previous`, `poutcome` features in the *active party*. Features `default`, `balance` are seen in *passive parties* 1 and 2, while `age`, `job`, `marital`, `education` are kept in *passive parties* 3 and 4.

**Adult Income Dataset:** We keep features `workclass`, `occupation`, `capital-gain`, `capital-loss`, `hours-per-week` in the active party and `race`, `marital-status`, `relationship`, `age gender`, `native-country` are kept by *passive parties* 1 and 2, while `education` is held by *passive parties* 3 and 4.

**Taobao Dataset:** We keep `pid`, `cms_group_id`, `cate_id`, `brand`, `new_user_class_level` , `price` features in the active party and `final_gender_code`, `age_level`, `occupation` are possessed by *passive parties* 1 and 2, while `pvalue_level`, `shopping_level` are kept in the *passive parties* 3 and 4.

**EMNIST:** The size of images in the dataset is $28 \times 28$. To emulate VFL, each image is equally partitioned into 4 "slices", sized $7 \times 28$, and held by 4 different parties, i.e., one active party and three passive parties.

**Fashion-MNIST:** The image size of this dataset is the same as that of EMNIST, and hence we adopt the same approach to partitioning the dataset.

**Model Architecture.** We consider different model architectures for each dataset. Features and models are partitioned among different parties in experimental settings. For the Banking dataset, the active party used Linear(57, 64); passive party 1 and 2 used unbiased Linear(3, 64); passive party 3 and 4 used unbiased Linear(20, 64). The three local modules combined are equivalent to Linear(80, 64). The global module owned by the aggregator comprised Linear(64, 1). For the Adult Income dataset, the active party, passive party 1 and 2, and passive party 3 and 4 possessed Linear(27, 64), unbiased Linear(63, 64), and unbiased Linear(16, 64) respectively. The three are equivalent to Linear(106, 64). The global module had Linear(64, 1). For the Taobao dataset, Linear(197, 128), Linear(11, 128), and Linear(6, 128), which were equivalent to Linear(214, 128), were utilized by the active party, passive party 1 and 2, and passive party 3 and 4 respectively. The aggregator maintained a global module with Linear(128, 1). We conduct experiments with CNN/MLP models on EMNIST/FMNIST datasets. For CNN on EMNIST, each passive party has Conv3x3 with 32 channels followed by Conv1x1 with 64 channels; each conv layer is followed by BatchNorm, ReLU, and MaxPool2x2; the active party has Linear(600, 120) and Linear(120, 47). For MLP on EMNIST, each passive party has Linear(196, 32) and Linear(32, 128); the active party has Linear(128, 256), Linear(256, 128), Liner(128, 64), and Linear(64, 47). The CNN and MLP used for FMNIST are the same as those for EMNIST; only the number of classes differs.

**Overhead profiling.** To quantify the additional communication and computation expenses associated with our method, we measured the total CPU time and the total amount of upload/ download bytes, including their respective overheads, compared to the unsecured split learning for VFL. In practice, parties should create new key pairs routinely, which mitigates the risk of adversaries potentially accessing all confidential information in the event of secret keys used in some iterations being unexpectedly leaked. In experiments, the key pairs and the shared secrets will be regenerated for every 5 iterations.

We used a learning rate of 0.01 and a batch size of 256. We applied ReLU activation to all layers except the output layer.

## A.2   PROOF FOR THE LEMMA

*Proof.* One thing we would like to make clear is that everything related to the cryptography is based on the positive integers. Therefore, before adding masks, we need to perform quantization to the intermediate output or gradients and then make them into the positive integers space, and everything is operated in the space of the integers modulo R ($Z_R$).

According to distributive law, we have:

$$(a \mod R + b \mod R) \mod R = (a + b) \mod R \tag{5}$$

The number R is chosen that both the sum of intermediate output or gradient and the mask generated by the PRG is in the integer space $Z_R$.

We can prove by induction on the number of clients (N) in the system.

First, assume there are 2 clients in the system: client 1 and client 2. We let the mask generated by the $\mathrm{PRG}(ss_{12})$ be $M$, which is uniformly distributed in the integer space $Z_R$. For a given intermediate output value $v_1$ and $v_2$ for client 1 and 2, the output after the mask is $(v_1 + M)$ and $(v_2 - M)$, so according to the distributive law, the results $(v_1 + M)$ and $(v_2 - M)$ are both uniformly distributed in $Z_R$.

Then, by induction and distributive law, with more clients in the system, since we are adding more uniformly distributed masks into the equation, the results are still uniformly distributed in $Z_R$. □

## A.3 ADDITIONAL RESULTS

Table 4 and 5 provides additional results.

| | Active Party CPU time (s) | | | | Passive Party CPU time (s) | | | |
| | Training phase | | Testing phase | | Training phase | | Testing phase | |
| Dataset | Total | Overhead | Total | Overhead | Total | Overhead | Total | Overhead |
|---|---|---|---|---|---|---|---|---|
| **EMNIST(Ours)** | **0.5551** | **0.3282** | **0.4786** | **0.3277** | **0.4000** | **0.1949** | **0.3423** | **0.1949** |
| EMNIST(SEAL) | $1.207 \times 10^3$ | $1.207 \times 10^3$ | $1.207 \times 10^3$ | $1.207 \times 10^3$ | $1.207 \times 10^3$ | $1.207 \times 10^3$ | $1.207 \times 10^3$ | $1.207 \times 10^3$ |
| EMNIST(PHE) | $1.757 \times 10^4$ | $1.757 \times 10^4$ | $1.757 \times 10^4$ | $1.757 \times 10^4$ | $1.757 \times 10^4$ | $1.757 \times 10^4$ | $1.757 \times 10^4$ | $1.757 \times 10^4$ |
| **FMNIST(Ours)** | **0.5438** | **0.3304** | **0.4723** | **0.3301** | **0.3947** | **0.2003** | **0.3338** | **0.2003** |
| FMNIST(SEAL) | $1.207 \times 10^3$ | $1.207 \times 10^3$ | $1.207 \times 10^3$ | $1.207 \times 10^3$ | $1.207 \times 10^3$ | $1.207 \times 10^3$ | $1.207 \times 10^3$ | $1.207 \times 10^3$ |
| FMNIST(PHE) | $1.757 \times 10^4$ | $1.757 \times 10^4$ | $1.757 \times 10^4$ | $1.757 \times 10^4$ | $1.757 \times 10^4$ | $1.757 \times 10^4$ | $1.757 \times 10^4$ | $1.757 \times 10^4$ |

Table 4: Results of MLP on the CPU time (in seconds)

| | Active Party Data Transmission (MB) | | | | Passive Party Data Transmission (MB) | | | |
| | Training phase | | Testing phase | | Training phase | | Testing phase | |
| Dataset | Total | Overhead | Total | Overhead | Total | Overhead | Total | Overhead |
|---|---|---|---|---|---|---|---|---|
| **EMNIST(Ours)** | **1.63** | **0.21** | **1.01** | **0.21** | **1.38** | **0.05** | **0.75** | **0.05** |
| EMNIST(SEAL) | 16.70 | 15.27 | 16.07 | 15.27 | 16.44 | 15.12 | 15.82 | 15.12 |
| EMNIST(PHE) | 137.26 | 135.83 | 136.63 | 135.83 | 137.01 | 135.68 | 136.38 | 135.68 |
| **FMNIST(Ours)** | **1.63** | **0.21** | **1.01** | **0.21** | **1.38** | **0.05** | **0.75** | **0.05** |
| FMNIST(SEAL) | 16.70 | 15.27 | 16.07 | 15.27 | 16.44 | 15.12 | 15.82 | 15.12 |
| FMNIST(PHE) | 137.26 | 135.83 | 136.63 | 135.83 | 137.01 | 135.68 | 136.38 | 135.68 |

Table 5: Results of MLP on the communication both in size (MB).

