# OpenReview forum: "vFedSec: Efficient Secure Aggregation for Vertical Federated Learning via Secure Layer"
_ICLR.cc/2024/Conference — ICLR 2024 Conference Withdrawn Submission_

### Official Review · Reviewer_8n1Z · 2023-10-17

**Soundness:** 2 fair
**Presentation:** 2 fair
**Contribution:** 2 fair
**Rating:** 3
**Confidence:** 4

**Summary:**

The authors propose a so-called "secure layer" for a generalized vertical FL (VFL) setting that allows clients to aggregate intermediate outputs in a privacy-preserving way. The protocol is based on a "masking with noise" approach, where clients first establish pairwise secret keys/seeds such that they can generate random masks that cancel out during aggregation. The authors evaluate the computation and communication overhead of their solution and compare it to two HE-based aggregation mechanisms, showing significant better performance.

**Strengths:**

The proposed solution (masking/noise approach for secure aggregation) is technically simple and easy to follow.
There are some attempts made to formally argue about security.
The performance is empirically compared to some alternative solutions.

**Weaknesses:**

First of all, it is a bit unclear how the secure layer fits into the generalized VFL setup. For this it would be helpful to first properly define the insecure generalized VFL version in §2.1, then explain which parts are replaced with secure aggregation. Figure 1 is not particularly helpful in this respect. Adding an additional figure illustrating the ideal functionality realized by the secure layer would certainly help. Furthermore, it might be helpful to first give a high-level overview as in §4 before going into the technical details presented in §3.

The proposed masking-based protocol looks like it's pretty much identical to what was originally proposed by Bonawitz et al. for their SecAgg protocol. While the authors mention that they adapt this idea, the differences and own contributions are unclear. It is also unclear whether other of the many secure aggregation mechanisms that have been proposed in FL could be utilized to implement the VFL secure layer use case. If so, the authors should have compared performance to other secure aggregation protocols instead of two extremely naive HE implementations.

Many works in VFL, as the authors also point out, consider only a two-party case with one active and one passive party. While it's great to have a generalized solution, it is unclear what the current practical relevance of the proposed solution (including the clustering approach) is. Some additional motivation would be great.

The described threat model (besides saying that both clients and the server are semi-honest) is a bit vague => are there additional non-collusion assumptions as, e.g., the label for each sample is valuable IP that is sent from the active party to the server should not be leaked to the passive parties. However, the authors claim  robustness against collusion between the aggregator and passive parties.

**Questions:**

- Could any secure aggregation approached be utilized to implement the secure layer instead of SecAgg?
- Are there any contributions in addition to using SecAgg?
- Could variants of SecAgg be adapted that have better asymptotic complexity and/or support drop-outs?
- Can you comment on robustness against collusion between the aggregator and passive parties considering that labels are apparently shared in the clear?

---

### Official Review · Reviewer_aMR8 · 2023-10-30

**Soundness:** 2 fair
**Presentation:** 2 fair
**Contribution:** 2 fair
**Rating:** 3
**Confidence:** 4

**Summary:**

Vertical Federated Learning (VFL) enables the computation of ML models from datasets where features of the same data points are distributed among different parties. Given its decentralized nature, it provides the flexibility to perform computations over data that is not desired to be shared, either for its communication cost or privacy reasons. However, VFL techniques either do not preserve privacy or resort into expensive cryptographic primitives such as Homomorphic Encryption to offer certain privacy guarantees.

This work explores the integration of a popular privacy enhancing primitive to VFL: Secure Aggregation. It shows that the combination of VFL and Secure Aggregation improves communication and computational cost compared to previous privacy preserving solutions without accuracy degradation.

**Strengths:**

Given its efficiency, secure aggregation is a natural candidate for privacy preserving federated learning. Therefore, exploring ways to integrate this primitive to VFL is an interesting research direction.

**Weaknesses:**

The paper has the following weaknesses:

1- Lack of novelty: the contribution presents a combination of VFL and Secure Aggregation. Even if I think that this is a nice idea, I do not see sufficient novelty on the contribution. Combining Secure Aggregation a FL setting without other added value is quite common in literature of privacy preserving techniques. Substantial parts of Section 3, 4 and 5 of the main text are devoted to explain already existent ideas present in the (already classic) Secure Aggregation technique of (Bonawitz, 2017). This kind of content is more appropriate to preliminary sections or appendices rather than main contribution sections.

2- Significance: the presented protocol shows good accuracy and communication cost, but it is unclear how much privacy is offered. Since the Secure Aggregation layer only protects an intermediate output, it is not clear how much information is disclosed.

3- Soundness:

3a- Insufficient security claims: the security analysis of a novel technique requires a more elaborate treatment, even in in the absence of no actively malicious adversaries (see for example [1] on how to prove security properties of multiparty protocols). Such analysis is present in (Bonawitz, 2017), but not in Section 5 + Appendix 2 of this contribution.

3b- Poor evaluation: it is not clear if the comparison with other privacy preserving techniques is fair. A fair comparison should also show how much privacy is offered by each technique. Some techniques might be more expensive, but they could also provide more privacy.

[1] https://eprint.iacr.org/2016/046

**Questions:**

Please address the significance (point 2) and evaluation (3b) points described above.

---

### Official Review · Reviewer_XS2L · 2023-10-31

**Soundness:** 3 good
**Presentation:** 2 fair
**Contribution:** 1 poor
**Rating:** 5
**Confidence:** 3

**Summary:**

This paper studies privacy-preserving federated learning (FL) in a setting where training data is vertically rather than not horizontally partitioned across clients. To do so, it introduces a so-called "secure layer" in the standard process for vertical FL that uses a masking-based/MPC approach similar to practical secure aggregation (Bonawitz et al.) to protect privacy in local computations. It implements the proposed protocol and compares it with two naive implementations based on homomorphic encryption, empirically showing the significant performance gain in terms of computation/communication overhead.

**Strengths:**

- This paper considers an important privacy problem in VFL.

**Weaknesses:**

W1. Major weakness of this paper is the novelty. Main contribution of this paper is to introduce a ""secure layer" to protect the privacy of local update of clients in VFL. It seems that the authors simply apply the existing (and very well known) secure aggregation protocol to VFL setting.

W2. This paper does not include recent works which have been studied privacy-preserving in VFL, and this paper compares the proposed protocol only to HE which is known to have large communication/computation overhead. For instance, recent works such as FedVS [1] and FedV [2], should be discussed and compared to highlight the contribution of this paper.

W3. This paper does not consider a dropout / straggler problem, which is a very common and important issue in practical FL.

W4. This paper has some typos, e.g.,
  - In page 2, $y=[L]$ => $y=[C]$?
  - In section 4, it says secure layer "will be" explained in Section 3. It is already explained.


[1] Li, Songze, Duanyi Yao, and Jin Liu. "FedVS: Straggler-Resilient and Privacy-Preserving Vertical Federated Learning for Split Models." arXiv preprint arXiv:2304.13407 (2023).

[2] Xu, Runhua, Nathalie Baracaldo, Yi Zhou, Ali Anwar, James Joshi, and Heiko Ludwig. "Fedv: Privacy-preserving federated learning over vertically partitioned data." In Proceedings of the 14th ACM Workshop on Artificial Intelligence and Security, pp. 181-192. 2021.

**Questions:**

Please see the weaknesses.